# Carbon Nanotubes and Polydopamine Modified Poly(dimethylsiloxane) Sponges for Efficient Oil–Water Separation

**DOI:** 10.3390/ma14092431

**Published:** 2021-05-07

**Authors:** Wen Zhang, Juanjuan Wang, Xue Han, Lele Li, Enping Liu, Conghua Lu

**Affiliations:** 1School of Materials Science and Engineering, Tianjin University, Tianjin 300072, China; zhangjike445@tju.edu.cn (W.Z.); lelelee86@163.com (L.L.); epliu@tju.edu.cn (E.L.); 2Tianjin Key Laboratory of Building Green Functional Materials, School of Materials Science and Engineering, Tianjin Chengjian University, Tianjin 300384, China; hanxue@tcu.edu.cn

**Keywords:** composite CNT-PDMS sponge, PDA modification, superhydrophilic/superoleophilic, oil–water separation, demulsification

## Abstract

In this paper, effective separation of oil from both immiscible oil–water mixtures and oil-in-water (O/W) emulsions are achieved by using poly(dimethylsiloxane)-based (PDMS-based) composite sponges. A modified hard template method using citric acid monohydrate as the hard template and dissolving it in ethanol is proposed to prepare PDMS sponge composited with carbon nanotubes (CNTs) both in the matrix and the surface. The introduction of CNTs endows the composite sponge with enhanced comprehensive properties including hydrophobicity, absorption capacity, and mechanical strength than the pure PDMS. We demonstrate the successful application of CNT-PDMS composite in efficient removal of oil from immiscible oil–water mixtures within not only a bath absorption, but also continuous separation for both static and turbulent flow conditions. This notable characteristic of the CNT-PDMS sponge enables it as a potential candidate for large-scale industrial oil–water separation. Furthermore, a polydopamine (PDA) modified CNT-PDMS is developed here, which firstly realizes the separation of O/W emulsion without continuous squeezing of the sponge. The combined superhydrophilic and superoleophilic property of PDA/CNT-PDMS is assumed to be critical in the spontaneously demulsification process.

## 1. Introduction

The frequent occurrence of oil spills and chemical leakage has addressed widespread attention due to its serious impact on both environment and human health. Therefore, many efforts have been devoted to oil–water separation [1,2], which can be generally classified into mechanical physical [3,4], chemical [5,6], and bioremediation methods [7,8,9,10]. Among them, physical absorption with lightweight porous oil absorbent is emerging as one of the most promising strategies as it is highly efficient, environmental-friendly, and easily operating. Many natural and synthetic absorbents materials such as activated carbon, cotton fiber, organic/inorganic hybrids, carbon nanotubes, and cross-linked polymers have since been employed in the form of meshes, textiles, foams/sponges, and aerogels [11,12,13]. Despite their good performances in oil absorption and removal, some weaknesses still lie in a majority of those materials, such as complex fabrication process, deficiency of oil recovery, poor durability and recyclability.

Recently, polydimethylsiloxane (PDMS) sponges are particularly attractive because of their excellent characteristics including intrinsic hydrophobicity/lipophilicity, high porosity and flexibility, good thermal and chemical stability, and easy fabrication, etc. [14,15,16,17]. Further, 3D interconnected PDMS sponges have been successful fabricated by a facile sacrificial template method, using sugar or NaCl as the hard template and water as the solvent [18,19]. Although presenting good adsorption performance, this strategy is troubled by the difficult and incomplete remove of the hard template, since water hardly penetrates into the internal pores of hydrophobicity PDMS. Yu et al. recently prepared PDMS sponges with efficient oil/water separation and stable recyclability by replacing sugar particles with citric acid monohydrate (CAM) as a hard template and dissolving it in ethanol [20]. Benefiting from the superwetting property of ethanol on PDMS, CAM could be easily removed in a shorter time, giving PDMS sponges with high porosity.

The porosity of PDMS sponges plays a key role in achieving high oil absorption. However, as the porosity increases, sponges with high adsorption capacity usually exhibit lower mechanical strength, which immensely impedes the practical application of PDMS sponges [21]. In order to improve the mechanical strength of PDMS sponges in a low cost, a simple surface modification by adding carbon nanomaterials (such as graphene sheets [22], carbon nanotubes (CNTs) [23,24,25,26], and carbon nanofiller [27]) with a hard template of the PDMS skeleton has been proposed. Despite the enhanced mechanical stability, carbon nanomaterials added by surface deposition are easy to fall off through practical operations such as ultrasound [26]. Thus, from a practical standpoint of oil–water separation, there is still an urgent demand for a low-cost, simple, and robust method to prepare porous PDMS sponges with considerable absorption capacity, stable performance, and high reusability.

On the other hand, oil sewage can be further classified into immiscible mixture, unstable dispersion, and stable emulsion, according to diameter (d) of the dispersed phase [28]. In terms of emulsion separation, emulsion droplets (d < 20 µm) usually have a large size mismatch with sponge macropores (100–1000 µm), which significantly limits the separation ability of absorbent sponge materials. Superhydrophobic modification of the sponge surface and the framework is an efficient strategy to improve the emulsion separation capacity, assisted by the combination of low surface energy chemicals and micro-nanoscale roughness [28,29,30,31]. For example, Liu et al. prepared a superhydrophobic oil–water separation melamine-formaldehyde (MF) sponge [28]. It contained a MF sponge matrix, a magnetic polydopamine (PDA) coating, and branched PDMS brushes through dopamine-mediated surface-induced atom transfer radical polymerization. Decorative PDMS brushes endow the sponge with effective separation of both water-in-oil (W/O) and oil-in-water (O/W) emulsions. However, due to the superhydrophobic nature of the sponge, external force is required to continuously squeeze the liquid into the sponge to achieve the separation of oil-in-water emulsions. Consequently, the development of new sponge materials that can separate O/W emulsions without the external assistance process is highly desired for the practical application.

To address the above remaining issues, we report the fabrication of functional composite sponges through modifying porous PDMS with CNTs and/or PDA via a facile and scalable hard template method, where CAM and ethanol are used as the hard template and template remover, respectively. In contrast, with the former reported surface-modification using CNTs, here CNT-PDMS composites are prepared by directly mixing non-curing PDMS with CNT dispersion before adding of the hard template. Thus, CTNs can be homogeneously introduced to both the surface and the matrix of porous PDMS, endowing the composite sponge with excellent hydrophobicity/lipophilicity, considerable absorption capacity, satisfactory mechanical strength, and high reusability. The as-prepared CNT-PDMS sponge exhibits highly efficient removal of oil from immiscible oil–water mixtures. It is notable that not only does it succeed in a bath absorption, the mechanical durable CNT-PDMS composite can also achieve continuous oil–water separation under both static and turbulent flow conditions, which gives it an attractive potential for large-scale industrial oil–water separation. Moreover, in order to effectively separate O/W emulsions without external compression, a PDA modified CNT-PDMS is prepared by simply dipping it into dopamine solution to accomplish the dopamine self-polymerization on the surface of CNT-PDMS. The combined superhydrophilic and superoleophilic property of PDA/CNT-PDMS plays a critical role in the separation process of O/W emulsion without continuous squeezing of the sponge, where the demulsification efficiency can be greatly improved by mechanical stirring of the emulsion. To our best knowledge, it is the first report for separating O/W emulsion without external continuous compression.

## 2. Materials and Methods

### 2.1. Materials

PDMS prepolymer and curing agent (Sylgard 184), and silicone oil (50CS) were purchased from Dow Corning (Midland, MI, USA). Carbon nanotubes (CNTs) with diameter 5–15 nm and length 10–30 mm (TNM1) were provided by TimesNano (Chengdu, China). Cyclohexane, n-hexane, toluene, tetrahydrofuran, tetrachloromethane, and ethanol were purchased from Tianjin Kemeier Chemical Reagent Co., Ltd. (Tianjin, China). Citric acid monohydrate (CAM), Sudan III, dopamine hydrochloride, and tris(hydroxymethyl)aminomethane (Tris) were purchased from Tianjin Ciens Biochemical Technology Co., Ltd. (Tianjin, China). NaOH, HCl, and cetyltrimethylammonium bromide (CTAB) were purchased from Sinopharm Chemical Reagent Co., Ltd. (Shanghai, China). All reagents were used as received.

### 2.2. Preparation of Porous PDMS Sponges

First, 5 g of PDMS prepolymer and curing agent in a ratio of 10:1 (*w/w*) were placed into a petri dish and diluted with 3 mL cyclohexane. Then, 30 g CAM was added as the hard template and fully mixed. After degassed for 1 h, the mixture was cured at 80 °C for 2 h to complete the polymerization. Finally, the sample was placed in ethanol for 6 h to remove CAM and cyclohexane, followed by oven drying at 40 °C. There were three CAM with different particle sizes: raw CAM (r-CAM), ground CAM (g-CAM), and mixed CAM (m-CAM), having r-CAM and g-CAM in a ratio of 1:1 (*w/w*). Unless otherwise specified, m-CAM was used in the following experiments.

### 2.3. Preparation of CNT-PDMS Sponges

Porous CNT-PDMS sponges were prepared similar to that described above for the PDMS sponge, except replacing 3 mL cyclohexane with 3 mL CNT dispersions in cyclohexane. The latter with different concentrations (e.g., 0.33, 1.67, 3.33, and 6 mg mL^−1^) were obtained by dispersing CNT in cyclohexane with different *w/v* ratio for 120 min using an ultrasonic breaker to ensure CNT was evenly distributed. Unless otherwise specified, 6 mg mL^−1^ CNT dispersion was used in the following experiments.

### 2.4. Preparation of PDA/PDMS and PDA/CNT-PDMS Sponges

First, dopamine solution was prepared by dissolving dopamine hydrochloride in 10 mM Tris-HCl buffer solution, and subsequently dipping NaOH into the solution carefully to adjust pH condition to 8.5 [32]. Then, the PDA/PDMS sponge and PDA/CNT-PDMS sponge were prepared by immersing the as-obtained PDMS sponge and CNT-PDMS sponge into the self-polymerization system of dopamine solution for 24 h, respectively. Finally, the PDA-coated sponges were thoroughly washed by water and ethanol, followed by oven drying at 40 °C.

### 2.5. Oil Absorption Experiments

Absorption capacities of different PDMS sponges for the following oils were measured: cyclohexane, n-hexane, toluene, tetrahydrofuran, tetrachloromethane, and silicone oil. Sponges cut into cubic shape (1 cm × 1 cm × 1 cm) were immersed in oils for 1 min and then picked out for weight measurements. All experimental processes were carried out at room temperature. The mass absorption capacities of oils were calculated by the following equation:(1)$$Absorption\;capability\;\left(\%\right)=\left(m_{1}-m_{0}\right)/m_{0}\times 100$$
where *m*_0_ and *m*_1_ are the weights of the sample before and after absorption, respectively.

In the oil–water separation experiments, immiscible oil–water mixtures were prepared by simply mixing water and oils (cyclohexane or tetrachloromethane stained with Sudan III). The used O/W emulsions were prepared by mixing toluene and water with the volume ratio of 1:99, where CTAB (1 mg mL^−1^) was used as an emulsifier. Typically, the oil was added dropwise into the water under vigorous stirring conditions, and then the mixture was vigorously stirred for 6 h to form a stable milky emulsion.

### 2.6. Characterization

The morphological and composition characterization was conducted using scanning electron microscope (SEM, Hitachi S-4800) (Tokyo, Japan) equipped with energy disperse spectroscopy (EDS) (Tokyo, Japan). The contact angle experiments were carried out using a Powereach contact angle goniometer (JC2000D) (Shanghai, China), equipped with a CCD camera (Shanghai, China). The mechanical measurements were conducted on a universal tensile testing machine (WDW-05, Jinan Shidai Group Co., Ltd., Jinan, China) with a 500 N load cell at room temperature. The compression tests of the sponges with dimensions of 1 × 1 × 1 cm^3^ were performed with a strain rate of 50 mm min^−1^. There was no interval between each cycle for cyclic loading tests. 

## 3. Results and Discussion

### 3.1. Preparation of PDMS-Based Sponges

Figure 1 shows the schematic procedure to prepare porous CNT-PDMS and PDA/CNT-PDMS sponges by a modified hard template method. First, non-curing PDMS sponges are mixed with CNT dispersion in cyclohexane, and then commercially available CAM particles are added and mixed. After thermal curing and removal of the CAM, CNT-PDMS sponges with interconnected pores are obtained. For the preparation of PDA/CNT-PDMS, the subsequently dipping step of CNT-PDMS in dopamine solution brought about PDA deposition on its surface, thereby forming PDA/CNT-PDMS. Besides, PDMS and PDA/PDMS sponges are prepared as well for comparative experiments.

### 3.2. Morphology and Wettability Characterization of PDMS-Based Sponges

Figure 2(a_1_–d_1_) shows the SEM image of PDMS, PDA/PDMS, CNT-PDMS, and PDA/CNT-PDMS sponges, respectively. All of the as-prepared sponges consist of the PDMS-based skeleton and the air gaps, which form the 3D interconnected pore structures. For template-assisted synthesis of porous PDMS, the size of the template can significantly influence the porosity and thus the oil absorption capacity [19,33].

Here, three different CAM templates are used, i.e., raw CAM (r-CAM) with an average particle size of ~800 μm (size distribution from 200 to 1300 μm, Appendix A), ground CAM (g-CAM) with an average particle size of ~100 μm (size distribution from 50 to 300 μm, Appendix A), and mixed CAM (m-CAM) having r-CAM and g-CAM in a ratio of 1:1 (*w/w*). It can be seen that the obtained PDMS sponge using r-CAM (Appendix A) has much larger pore sizes than that using g-CAM (Appendix A), while their counterpart using m-CAM has both large and small pores (Figure 2(a_1_)). It is reported that the size-mixed hard template usually gives rise to the higher porosity than relatively uniform sugar cases, leading to larger absorption capacity [17], which is also identified later in this paper. Thus, m-CAM is utilized in the preparation of the following PDMS-based sponges (Figure 2(b_1_–d_1_)).

CNT-PDMS sponges were prepared by mixing CNTs (Appendix A) into the PDMS matrix. The presence of CNTs in the porous materials has two critical roles. On the one side, they are used as the reinforcing filler to improve the mechanical property of sponges (discussed in detail later). On the other side, the introduction of CNTs in the composite sponge can contribute to the enhancement of hydrophobicity, also a key element to absorption capacity besides porosity [25]. The water contact angle of CNT-PDMS (137.9°, inset in Figure 2(b_1_)) is higher relative to that of pure PDMS (127.8°, inset in Figure 2(a_1_)), indicating that the presence of CNTs increases the hydrophobicity. The smooth surface of porous PDMS (Figure 2(a_2_)) is evidently roughened by the presence of CNTs (Figure 2(b_2_)). In addition, the increase in carbon element lowers the surface energy for the CNT-PDMS. The synergism of rougher surface and smaller surface energy leads to the improved hydrophobicity [24].

PDA has excellent hydrophilic/oleophilic. This distinct characteristic endows PDA-modified PDMS sponges with an unexpected advantage in the separation of oil-in-water emulsions without the assistance of external compression (see details later). Figure 2(c_1_,d_1_) are SEM images of PDMS and CNT-PDMS sponges after the deposition of PDA, respectively, which show rougher surface compared to their corresponding precursors. The yellow appearance of PDA/PDMS (Appendix A) verifies the successful deposition of PDA onto the surface of porous PDMS. The element distribution of PDA/CNT-PDMS is shown in Appendix A, where the presence of N element proves that PDA has deposited on the CNT-PDMS sponge as well. With the introduction of PDA, the surface of PDA/PDMS and PDA/CNT-PDMS transform into super-hydrophilicity (water contact angle <5°, inset in Figure 2(c_1_,d_1_)), which can be ascribed to the cooperation effect of hydrophilic PDA and high roughness from the porous structure [34].

Although having different water wettability, all of the above mentioned four kinds of sponges show superoleophilic properties. Once droplets of example oil (e.g., cyclohexane) were placed on the surface of sponges, they were immediately absorbed into the 3D porous structure, resulting in an oil contact angle of nearly 0° (Appendix A). Bumps on the surface of sponges occurred just following the absorption (Appendix A). This provides evidence of the swellable nature of all prepared sponges, which is further confirmed by the significant volume increase after the sponges entirely soaked into cyclohexane (Appendix A).

### 3.3. Performance of Oil Absorption for PDMS-Based Sponges

All the porous PDMS-modified sponges have the ability to quickly take up oil as shown above. In order to verify the contribution of CNTs in the oil absorption and thus select the optimal condition, different samples are prepared by using the r-CAM template and different CNTs concentrations (0.33, 1.67, 3.33, and 6 mg mL^−1^), which correspond to different ratios (0.02%, 0.1%, 0.2%, and 0.36% wt, respectively) of CNTs with respect to the PDMS matrix. From Figure 3a, it can be seen that the absorption capacity for cyclohexane monotonically increases from ~700% to 900% with CNTs content from 0% to 0.36%. The same tendency exists for all the remaining tested oil, including n-hexane, toluene, tetrahydrofuran, tetrachloromethane, and silicone oil. Since the CNT solution with concentration exceeding 6 mg mL^−1^ almost becomes a paste and could not be uniformly dispersed in the PDMS matrix, CNT-PDMS sponge containing 0.36% CNTs was selected for the following research.

The size of the hard template is important for oil absorption as well, so pure PDMS sponges prepared by using CAM with different size distribution were investigated. It was found that absorption capacities of the porous PDMS using m-CAM were obviously improved relative to that using r-CAM (Appendix A). The data for PDMS sponge using g-CAM is not shown, as the prepared sponge is very soft and will collapse once absorbing oil. This may be because the sponge pores are too dense to support the PDMS skeleton itself after absorbing oil. Similar to pure PDMS, the absorption capacities of CNT-PDMS can be further enhanced when the template was changed from r-CAM to m-CAM (Figure 3b). In brief, 6 mg mL^−1^ CNT solution and m-CAM are the optimal condition for preparing porous PDMS-based sponges in this research.

Next, absorption capacities of four sponges (i.e., PDMS, PDA/PDMS, CNT-PDMS, PDA/CNT-PDMS) prepared with respective optimal condition were investigated. For all tested oils used here, the three modified PDMS sponges show enhanced absorption capacities relative to pure PDMS sponge (Figure 3c), regardless of the surface hydrophilic/hydrophobic. This may be attributed to the combination of superlipophilicity and enhanced surface roughness of all modified PDMS. CNT-PDMS and PDA/CNT-PDMS have almost similar and the highest absorption capacities among the four sponges, evidencing the positive role of CNTs in the oil absorption. The optimal sponge can achieve high absorption capacity not only for polar organic solvents (e.g., tetrahydrofuran) but also for nonpolar organic solvents and oils (e.g., cyclohexane) in the range from ~700 to 2500%, with the maximum absorption capacity reaching up to 25 times its weight. The difference in mass absorption capacities for organic solvents is mainly due to the density discrepancy, such as cyclohexane (0.78 g mL^−1^, 20 °C), n-hexane (0.66 g mL^−1^, 20 °C), toluene (0.87 g mL^−1^, 20 °C), tetrahydrofuran (0.89 g mL^−1^, 20 °C), and tetrachloromethane (1.60 g mL^−1^, 20 °C).

The recyclability and durability of oil absorbents are of significance in practical oil cleanup applications to reduce the cost. The absorbed oils in the CNT-PDMS and PDA/CNT-PDMS sponges can be removed by manually squeezing the sponge due to the high-elasticity nature of the PDMS skeleton. The recyclability and durability of PDMS-based sponges are evaluated by immersing them into cyclohexane repeatedly, followed by compression to remove the oil and drying. As shown in Figure 3d, no obvious decrease in the absorption capacity can be observed in the durable test of 10 recycles, indicating their excellent stability.

### 3.4. Mechanical Property of PDMS-Based Sponges

It is well known that PDMS materials are highly compressible. All the four PDMS-based sponges can endure a compression of more than 70% and return to its original state after removing the applied force (Appendix A). This outstanding resilience performance ensures the recyclability and durability of PDMS-based sponges mentioned above.

To further evaluate their mechanical property and investigate the effect of CNTs/PDA modification, the cyclic compressive experiments were conducted on a universal tensile testing machine (Figure 4a). The compressive stress–strain curves of PDMS, CNT-PDMS, PDA/CNT-PDMS sponges are presented in Figure 4b–d, respectively. First, all of them show a closed compression-release curve, corresponding to their excellent resilience performance. The existing hysteresis loops in curves indicate substantial energy dissipation, resulting from a combined contribution of the intrinsic viscoelasticity of PDMS and the extrusion of air from the open pore structure of the porous materials [35]. There is no significant decay in the compressive stress during 10 cycles at 80% strain, indicating that all three sponges have good mechanical durability.

Except the basic compressive property, the PDMS sponge before and after modification shows distinct difference in the stress value at the same strain. From a clearer comparison (Appendix A), CNT-PDMS and PDA/CNT-PDMS sponges have higher stress level compared with the pure PDMS during the whole range of the tested strain. In the strain of 80%, the stress of the PDA/CNT-PDMS sponge reaches 32 KPa, which is 3 times that of the pure PDMS. Besides, the 10 cycles of compression-release of the modified sponges hardly fluctuate under a large strain. Those results confirm that the modified PDMS sponge can withstand greater compressive stress, indicating the addition of CNT and PDA effectively strengthens the PDMS sponge and thus providing them as promising candidates for practical oil/water separation application.

### 3.5. Application in Selective Oil Absorption from Immiscible Oil–Water Mixtures

Thanks to the hydrophobic/superoleophilic property, combined with satisfactory mechanical strength, the porous CNT-PDMS sponge may be an ideal candidate as the oil absorbent for oil/water separation purposes. Herein, the applications of CNT-PDMS absorbent in selective batch absorption and continuous absorption of oil from immiscible oil–water mixtures are demonstrated (Figure 5).

During batch absorption, two different model organic solvents, cyclohexane and tetrachloromethane (both stained with Sudan III), are dropped into the water to demonstrate the selective absorption of CNT-PDMS (Figure 5(a_1_,a_2_,b_1_,b_2_)). Since the density of cyclohexane is smaller than that of water, an immiscible oil/water mixture with cyclohexane floating on the water surface is prepared (Figure 5(a_1_)). When placed on the surface of the as-prepared oil/water mixture, CNT-PDMS absorbs oil completely in ~20 s and its 3D interconnected pore structure allows the storage of the collected oil inside the sponge (Figure 5(a_2_) and Appendix A). Similarly, the heavier tetrachloromethane drop sunk at the bottom of the water (Figure 5(b_1_)) could also be quickly absorbed by the sponge in ~15 s (Figure 5(b_2_) and Appendix A). Furthermore, the CNT-PDMS sponge can easily remove organic solvents from water, while no water drops were absorbed due to its excellent hydrophobicity.

On the other hand, the CNT-PDMS absorbent can be used as a filter for continuous oil–water separation, while a vacuum pump was used as a power unit to explore the potential application in large-scale oil absorption. The continuous removal of oil (dyed cyclohexane used as a model) from water at the static or dynamic state is demonstrated in Figure 5(c_1_,c_2_) and Figure 5(d_1_,d_2_), respectively. Here, the r-CAM template is used for the preparation of CNT-PDMS sponges with bigger pores to lower the pressure requirement. The enhanced mechanical property benefiting from the introduction of CNTs plays a crucial function in the continuous separation, as the pure PDMS sponges could not bear the pumping pressure and thus become fragmented in the continuous separation experiments.

For the continuous absorption experiment at static state, a piece of CNT-PDMS sponge is partially squeezed in one end of a pipe, while the other end of the pipe is connected to a suction flask with an operating vacuum pump. Once placing the sponge at the oil/water interface, the oil is selectively removed from the water to the suction flask at a fast rate, leaving only water in the beaker (Figure 5(c_1_,c_2_) and Appendix A). Meanwhile, no water is observed to flow into the suction flask even after putting CNT-PDMS into water, which also proves that the porous CNT-PDMS has excellent hydrophobic properties.

Considering the actual environmental conditions such as lakes, rivers, or oceans, the water surface is always in motion and not calm as the previous experiments (Figure 5(c_1_,c_2_)). Therefore, magnetic stirring at the bottom of the beaker is utilized here to simulate the turbulent flow conditions under a real environment, which transform the immiscible oil/water mixture (inset in Figure 5(d_1_)) into numerous oil droplets in the water (Figure 5(d_1_)). As shown in Figure 5(d_2_) and Appendix A, the dyed cyclohexane droplets are completely pumped into the suction flask under turbulence condition as well as static condition. Similarly, the water remains and is not adsorbed by the CNT-PDMS sponge.

### 3.6. Application in the Separation of O/W Emulsions

Compared with immiscible oil–water mixtures, the separation of surfactant-stabilized emulsions (especially O/W type) still faces more severe challenges, which usually require high cost and complex treatments. For W/O emulsions, the hydrophobic absorbents can serve as a filter to separate the mixtures under gravity driving condition [29,30]. However, with respect to O/W emulsions, the hydrophobic nature of absorbents makes them repel water. Thus, consecutive compression of the sponges is always required to force oil in water droplets to penetrate the sponge pores. To deal with this issue, PDA modification is introduced to the initial hydrophobic sponge, which realizes the effective separation with no need of external force.

Here, a model O/W emulsion is prepared using toluene as the oil and CTAB (1 mg mL^−1^) as the emulsifier in water. The emulsion solutions appear white milky color (Figure 6(a_1_)) with numerous tiny toluene droplets (Figure 6(b_1_)). All four sponges of PDMS, CNT-PDMS, PDA/PDMS, and PDA/CNT-PDMS are respectively placed on the emulsion surface and kept for 50 min. It can be seen that PDA deposited sponges gradually clear the toluene droplets while the emulsion solutions with their counterparts stay milky color (Appendix A). Especially for PDA/CNT-PDMS, all the toluene droplets are absorbed, and the solution becomes transparent after 40 min (Figure 6(a_2_)). The O/W emulsion separation is further confirmed by the optical microscopy images (Figure 6(b_2_)), where no droplet can be observed.

The above experiment is conducted under static state. Similar to the demonstration for selective oil absorption from immiscible oil–water mixtures, a dynamic environment for demulsification is mimicked as well. As shown in Figure 6(c_1_), a piece of CNT-PDMS sponge and PDA/CNT-PDMS sponge are placed on the surface of the O/W emulsion under constant stirring. The milky emulsion with PDA/CNT-PDMS quickly becomes transparent in ~5 min (Figure 6(c_2_) and Appendix A), indicating that stirring can accelerate the demulsification process. In contrast, the emulsion with CNT-PDMS still presents a slower oil absorption even under stirring. These findings convince that the PDA/CNT-PDMS sponge can provide a promising alternative for O/W typed wastewater treatment.

The separation physics of O/W emulsions through PDA/CNT-PDMS is also proposed. Thanks to the superhydrophilic surface of the sponge deposited with PDA, oil drops wrapped by water can spontaneously enter the internal porous structures of the sponge without applying external force. Then, the hydrophobic/oleophilic CNT-PDMS skeleton acts effectively to trap tiny emulsified oil droplets, making the emulsion gradually clear. This can be evidenced by the result that PDA/CNT-PDMS has a better demulsification effect than PDA/PDMS. By comparison, both porous PDMS and CNT-PDMS exhibit hydrophobic surface structure, so the O/W emulsion cannot easily penetrate the sponge, resulting in unsatisfactory emulsion separation.

## 4. Conclusions

In summary, we have developed a revised hard template method to prepare porous PDMS-based sponges modified with CNTs and/or PDA. The CNT-PDMS composite sponge exhibits better comprehensive properties including hydrophobicity, absorption capacity, and mechanical strength than the pure PDMS sponge. The influence of the particle size of sacrificial template CAM and the CNTs content on oil absorption capacity is systematically investigated to find the optimized preparation condition. The CNT-PDMS sponge shows outstanding absorption performance for various oils and can be reused by extrusion and evaporation. Notably, assisted with vacuum pumps, continuous oil/water separation is further achieved regardless of a calm or turbulent water surface. On the basis of the CNT-PDMS composite, just as remarkable is that the effective separation of O/W emulsions without external compression is for the first time realized by modifying the CNT-PDMS surface with a layer of PDA. The separation mechanism is proposed, in which the combined superhydrophilic and superoleophilic property of PDA/CNT-PDMS is assumed to be critical in the spontaneously demulsification process.

## Figures and Tables

**Figure 1 materials-14-02431-f001:**
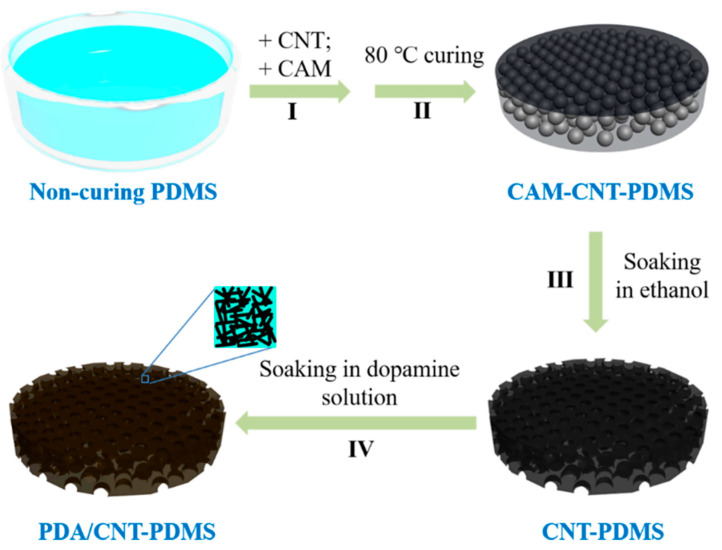
Schematic illustration of the preparation of CNT-PDMS and PDA/CNT-PDMS sponges.

**Figure 2 materials-14-02431-f002:**
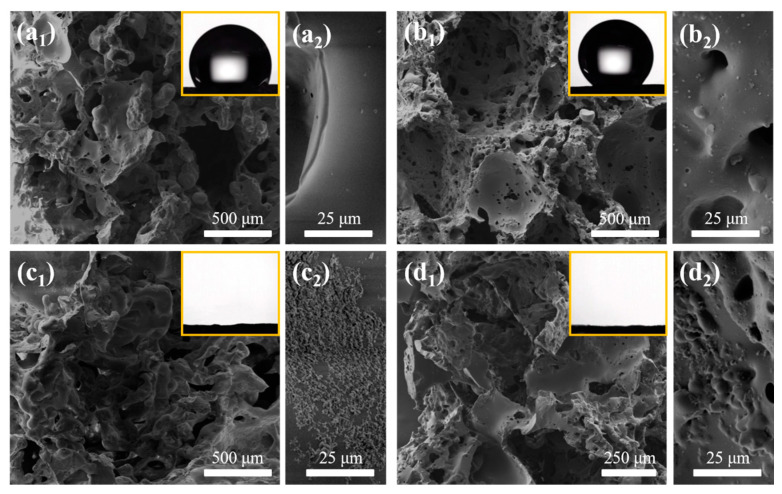
SEM images of porous PDMS (**a_1_**), CNT-PDMS (**b**_1_), PDA/PDMS (**c_1_**), and PDA/CNT-PDMS (**d_1_**) prepared with the m-CAM template. (**a_2_**–**d_2_**) are the corresponding zoomed SEM images. The inset in the upper-right corner of (**a_1_**–**d_1_**) shows the water contact angle on the corresponding sponge surface.

**Figure 3 materials-14-02431-f003:**
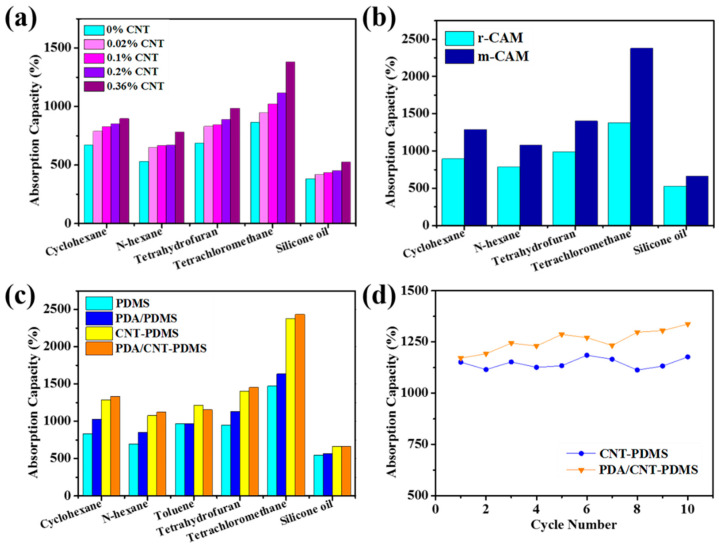
The influence of CNT content (**a**) and the particle size of CAM (**b**) on the absorption capacity of CNT-PDMS sponge for various organic solvents and oils. (**c**) The absorption capacity of PDMS, CNT-PDMS, PDA/PDMA, and PDA/CNT-PDMS sponges prepared with optimized condition for various organic solvents and oils. (**d**) Stability of CNT-PDMS and PDA/CNT-PDMS sponges with multi-recycles of cyclohexane absorption and removal.

**Figure 4 materials-14-02431-f004:**
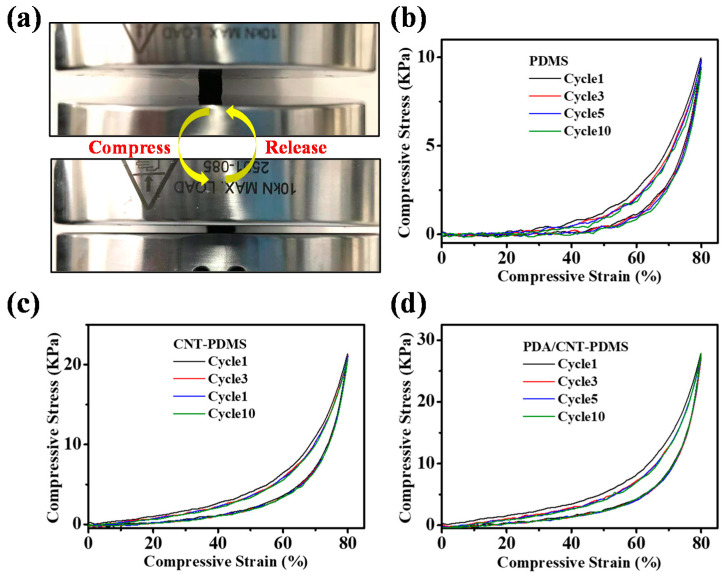
(**a**) Digital images of a uniaxial compressive test cycle. Compressive stress–strain curves of PDMS (**b**), CNT-PDMS (**c**), and PDA/CNT-PDMS (**d**) sponges at the maximum strain of 80% for 10 cycles.

**Figure 5 materials-14-02431-f005:**
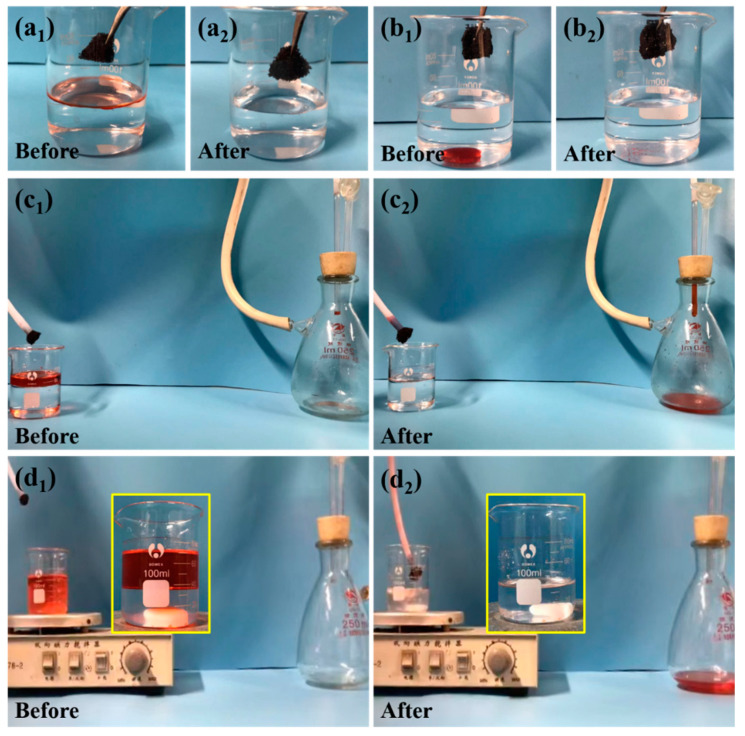
The use of CNT-PDMS sponge for batch absorption (**a_1_**,**a_2_**,**b_1_**,**b_2_**) of cyclohexane (**a_1_**,**a_2_**) and tetrachloromethane (**b**_1_,**b**_2_) from immiscible oil–water mixtures, and continuous absorption (**c_1_**,**c_2_**,**d_1_**,**d_2_**) of cyclohexane from immiscible oil–water mixtures under static (**c_1_**,**c_2_**) or dynamic (**d_1_**,**d_2_**) condition. (**a_1_**,**b_1_**,**c_1_**,**d_1_**) are photos of corresponding oil/water mixtures before contact with the CNT-PDMS sponge, and (**a_2_**,**b_2_**,**c_2_**,**d_2_**) are photos of corresponding oil/water mixtures after oil absorption. The inset in (**d_1_**) is the photo of the as-prepared oil–water mixture before magnetic stirring and vacuum pumping, and the inset in (**d_2_**) is the photo of the oil–water mixture after oil absorption.

**Figure 6 materials-14-02431-f006:**
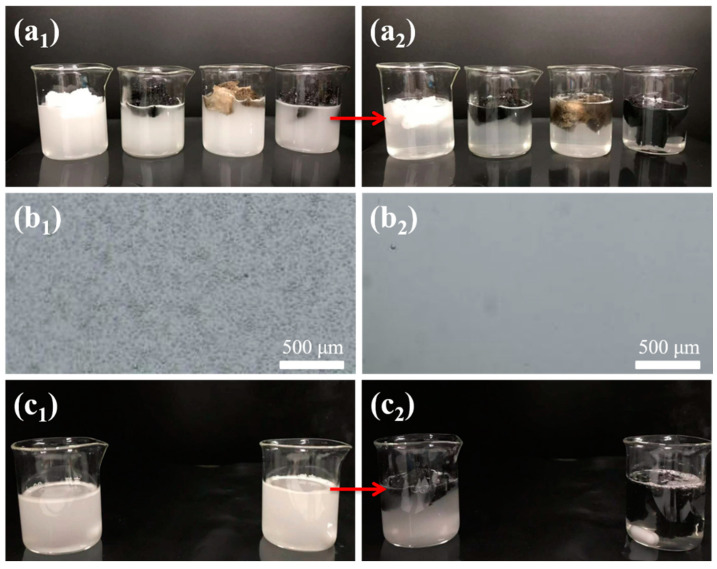
The separation of O/W emulsions under static (**a_1_**,**a_2_**) and dynamic (**c_1_**,**c_2_**) condition. The photos of PDMS, CNT-PDMS, PDA/PDMS, and PDA/CNT-PDMS sponges (from left to right) in toluene/water emulsion at 0 min (**a_1_**) and 40 min (**a_2_**), while (**b_1_**) and (**b_2_**) are corresponding optical images of emulsion. The photos of CNT-PDMS (left) and PDA/CNT-PDMS sponges (right) in toluene-in-water emulsion with magnetic stirring for 0 min (**c_1_**) and 5 min (**c_2_**).

## Data Availability

The data presented in this study are available on request from the corresponding author.

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
