# Peer review of "Carbon Nanotubes and Polydopamine Modified Poly(dimethylsiloxane) Sponges for Efficient Oil–Water Separation"

_materials, 2021, doi:10.3390/ma14092431_

Round 1

Reviewer 1 Report

The manuscript “Carbon Nanotubes and Polydopamine Modified Poly(dimethylsiloxane) Sponges for Efficient Oil–Water Separation” addresses an interesting topic and clearly describes the results.

Moreover, the paper also uses a good English. Therefore, it may be recommended for publication after minor revision:

1)   1 - Authors should improve the quality of the figures.

      Figures 2, 3, 4 (b) (c) (d)and 6 (b1) (b2) has a very low quality;

      Figures 4 (b) (c) (d) is unpresentable.

2)  2 - Figures need to be created with the same and unique template:

           the legends must be of the same size and visible (all figures);
     figure 5 is very confused; the size of the legend and the images could be reduced and improved.

3 - In the introduction part it is recommended to add some more discussion. The following publication is recommended to fulfill this section:

        Surface Modifications Induced in UHMWPE Based Nanocomposites during the Ageing in Simulated Synovial Fluid. In: Macromolecular Symposia. 2020. p. 1900055 

Author Response

Dear Jill Zhang,

Editor of Materials

We would like to thank you very much for your precious time and review of our manuscript. We also would like to thank the reviewers for their positive evaluation, approval of significant contribution of our manuscript, and valuable comments. Enclosed please find the revised version of our manuscript "Carbon Nanotubes and Polydopamine Modified Poly(dimethylsiloxane) Sponges for Efficient Oil–Water Separation". In the file “Response to Reviewers”, we provide a point-by-point response to the reviewers’ comments which contribute to improve our work. Uploaded files include the “Revised Manuscript” with all highlighted changes, “Revised Supporting Information” with all highlighted changes, a .zip archive containing the production data, and Response to Reviewers.

We believe that we have addressed all the concerns raised by the reviewers. We thank the reviewers again for the valuable feedback and your timely attention to this matter.

Yours sincerely,

Comments:

The manuscript “Carbon Nanotubes and Polydopamine Modified Poly(dimethylsiloxane) Sponges for Efficient Oil–Water Separation” addresses an interesting topic and clearly describes the results. Moreover, the paper also uses a good English. Therefore, it may be recommended for publication after minor revision:

Response: We thank the reviewer very much for the positive evaluation and approval of significant contribution of our manuscript. All the minor concerns raised by the reviewer have been addressed point-by-point in the following.

Note that the Page/Line numbers mentioned in the Response Letter are based on the files of PDF version named as “Revised Manuscript” and “Revised Supporting Information” with all highlighted changes. The mentioned figures named such as Figure. R1 are shown in the Response Letter. The mentioned figures named such as Figure. 1 and Figure. S1 are shown in the Revised Manuscript and Supporting Information.

Q1. Authors should improve the quality of the figures. Figures 2, 3, 4 (b) (c) (d)and 6 (b1) (b2) has a very low quality; Figures 4 (b) (c) (d) is unpresentable.

Response: Thank the reviewer for the helpful comment.

(1) The quality of pictures mentioned by the reviewer has been improved according to the requirements of the journal. As shown below, taking Figure 2(a1) for example, the resolution of the pictures has been improved from 598*516 to 1196*1032.

Then the last figures in Figure 2 have been modified in a similar fashion, and grouped as shown in Figure R1. With respect to Figure 3, Figure 4b,c,d and Figure 6b1,b2, they have been improved in the same way.

Please see Figure 2 on Page 6 Line 221, Figure 3 on Page 8 Line 272, Figure 4 on Page 10 Line 304, and Figure 6 on Page 13 Line 398 of the Revised Manuscript.

Besides, after careful proofreading, we are sorry that there is a misuse of Figure 6b1, which is the picture of toluene/water emulsion using SDS as emulsifier. It should be toluene/water emulsion using CTAB as emulsifier. The mistake has been corrected as following, and does not affect the existing conclusions.

(2) Figure 4b-d show uniaxial compressive tests of PDMS, CNT-PDMS, PDA/CNT-PDMS sponges, which were conducted on a universal testing machine. We guess there may be some ambiguous statements in our manuscript. Thus, we have added experimental details for the mechanical measurement in Part 2.6 “Characterization”, and revised relevant statement in Part 3.4 “Mechanical property of PDMS-based sponges”. Besides, the original Figure 4a1-a3 was moved to the Supporting Information, and replaced with digital images of the uniaxial compressive test (Figure 4a).

Please refer to Page 4 Line 155-159, Page 9 Line 284-285 and Figure 4a on Page 10 Line 304 of the Revised Manuscript, and Figure S8 on Page S-5 of the Revised Supporting Information.

Q2. Figures need to be created with the same and unique template; the legends must be of the same size and visible (all figures); figure 5 is very confused; the size of the legend and the images could be reduced and improved.

Response: Thanks a lot for the valuable suggestion and comment.

(1) All the figures and legends have been modified to the same size and template.

(2) In terms of Figure 5, it includes two types of separation experiments: (i) the selective batch absorption of oil from immiscible oil-water mixtures (Figure 5a1,a2,b1,b2), (ii) the selective continuous absorption of oil from immiscible oil-water mixtures at static (Figure 5c1,c2) or dynamic (Figure 5d1,d2) condition. For the selective batch absorption experiment, Figure 5a1 shows the initial immiscible cyclohexane/water mixture before contacting with the CNT-PDMS sponge, while Figure 5a2 shows the mixture after oil absorption. The absorption process can be seen in Movie S1. Figure 5b1 and b2 is the same as 5a1 and a2, except replacing cyclohexane with tetrachloromethane. The absorption process for the tetrachloromethane/water mixture can be seen in Movie S2. For the selective batch absorption experiment at static condition, Figure 5c1 shows the initial immiscible cyclohexane/water mixture before contacting with the CNT-PDMS sponge which is connected to a vacuum pump, while Figure 5c2 shows the mixture after oil absorption. The corresponding absorption process can be seen in Movie S3. The difference between Figure 5d1,d2 and c1,c2 is that a magnetic stirring at the bottom of the beaker is added in the former to simulate the dynamic environment, whose absorption process can be seen in Movie S4.

For better clarity, we have revised the caption of Figure 5 as follows: The use of CNT-PDMS sponge for batch absorption (a1,a2,b1,b2) of cyclohexane (a1,a2) and tetrachloromethane (b1,b2) from immiscible oil-water mixtures, and continuous absorption (c1,c2,d1,d2) of cyclohexane from immiscible oil-water mixtures under static (c1,c2) or dynamic (d1,d2) condition. (a1,b1,c1,d1) are photo pictures of corresponding oil/water mixtures before contacting with the CNT-PDMS sponge, and (a2,b2,c2,d2) are photo pictures of corresponding oil/water mixtures after oil absorption. The inset in (d1) is the photo picture of as-prepared oil-water mixture before magnetic stirring and vacuum pumping, and the inset in (d2) is the photo picture of oil-water mixture after oil absorption.

Relevant figures and statements have been revised in the revised manuscript, please refer to Figure 3 on Page 8 Line 272, Figure 4 on Page 10 Line 304, Figure 6 on Page 13 Line 398, and Page 11 Line 353-356 of the revised manuscript.

Q3. In the introduction part it is recommended to add some more discussion. The following publication is recommended to fulfill this section: Surface Modifications Induced in UHMWPE Based Nanocomposites during the Ageing in Simulated Synovial Fluid. In: Macromolecular Symposia. 2020. p. 1900055

Response: Thanks a lot for the suggestion. We have added more discussion based on the above publication as a new reference [27] in the introduction part, please see the Revised Manuscript on Page 2 Line 56-57 and Page 15 Line 501-502. 

Reviewer 2 Report

Article is very interesting. The resulting material works very well.

As for the presentation, I suggest moving the figures 4a, 5 and 6 to Supplemtary materials.

According to mechanical properties, there are specialized equipment for measuring mechanical properties. This is the weakest part of this work. 

Line 270-274 The compressibility test should be performed mechanically, similar to the tensile test. Performing the "measurement" by hand gives very unreliable results.

Fig. 4 b-d I do not understant, how these results were obtained. There is no information i Methods part.

Part 2.6 Characterization has to be improved to clearly present the procedures applied during the study. At this stage it is not clear.

Is there also an option, that I missunderstood the idea of Authors. However, at lest the Characterization part has to be desribet in more details to make all procedures clear. If yes, I can agree to  modified the decision to "minor revision"

Author Response

Dear Jill Zhang,

Editor of Materials

We would like to thank you very much for your precious time and review of our manuscript. We also would like to thank the reviewers for their positive evaluation, approval of significant contribution of our manuscript, and valuable comments. Enclosed please find the revised version of our manuscript "Carbon Nanotubes and Polydopamine Modified Poly(dimethylsiloxane) Sponges for Efficient Oil–Water Separation". In the file “Response to Reviewers”, we provide a point-by-point response to the reviewers’ comments which contribute to improve our work. Uploaded files include the “Revised Manuscript” with all highlighted changes, “Revised Supporting Information” with all highlighted changes, a .zip archive containing the production data, and Response to Reviewers.

We believe that we have addressed all the concerns raised by the reviewers. We thank the reviewers again for the valuable feedback and your timely attention to this matter.

Yours sincerely,

Comments: Article is very interesting. The resulting material works very well.

Response: We thank the reviewer very much for the positive evaluation and approval of significant contribution of our manuscript. The reviewer’s helpful comments have been considered in the revised version of our manuscript.

Note that the Page/Line numbers mentioned in the Response Letter are based on the files of PDF version named as “Revised Manuscript” and “Revised Supporting Information” with all highlighted changes. The mentioned figures named such as Figure. R1 are shown in the Response Letter. The mentioned figures named such as Figure. 1 and Figure. S1 are shown in the Revised Manuscript and Supporting Information.

Q1. As for the presentation, I suggest moving the figures 4a, 5 and 6 to Supplemtary materials.

Response: Thanks a lot for the helpful and valuable comment. We have moved Figure 4a1-a3 to Supporting Information, please see Figure S8c1-c3 on Page S-5 of the Revised Supporting Information. However, in terms of Figure 5 and 6, they exhibit the application of PDMS-based sponges in selective oil absorption from immiscible oil-water mixtures and O/W emulsions, respectively, which are the focus of this article. Especially, the separation for emulsion is the major innovation of this study. Therefore, it would be inappropriate if Figure 5 and 6 were moved to Supporting Information.

Besides, after careful proofreading, we are sorry that there is a misuse of Figure 6b1, which is the picture of toluene/water emulsion using SDS as emulsifier. It should be toluene/water emulsion using CTAB as emulsifier. The mistake has been corrected as following, and does not affect the existing conclusions.

Q2. Line 270-274 The compressibility test should be performed mechanically, similar to the tensile test. Performing the "measurement" by hand gives very unreliable results.

Response: Thank the reviewer for pointing out this. Actually, we performed the compression test instead of tensile test on a universal tensile testing machine. In Part 3.4 “Mechanical property of PDMS-based sponges”, firstly, the high elasticity of PDMS-based sponges was simply exhibited by hand compression/release to give an intuitive impression, which is described in Line 270-274 of the original manuscript. Then further compressive tests were conducted on a universal tensile testing machine, whose results were shown in Figure 4b-d.

We guess there may be some ambiguous statements in our manuscript, so that the reviewer misunderstood. Thus, we have added experimental details in Part 2.6 “Characterization”, and revised relevant statement in Part 3.4 “Mechanical property of PDMS-based sponges”. Besides, the original Figure 4a1-a3 was moved to the Supporting Information, and replaced with digital images of the uniaxial compressive test (Figure 4a).

Relevant parts have been revised in the revised manuscript, please refer to Page 4 Line 156-162, Page 9 Line 284-285 and Figure 4a on Page 10 Line 304 of the Revised Manuscript, and Figure S8 on Page S-5 of the Revised Supporting Information.

Q3. Fig. 4 b-d I do not understand, how these results were obtained. There is no information i Methods part.

Response: Thank the reviewer for pointing out this. Figure 4b-d shows the compressive stress-strain curves of PDMS, CNT-PDMS, and PDA/CNT-PDMS sponges, respectively, which are measured by a universal tensile testing machine. We have revised Part 2.6 Characterization to provide more specific information about the measurement procedure, please see Page 4 Line 155-159 of the Revised Manuscript.

Q4. Part 2.6 Characterization has to be improved to clearly present the procedures applied during the study. At this stage it is not clear.

Response: Thanks a lot for the helpful comment. We have revised Part 2.6 to provide a better presentation. The added part is as follows: The mechanical measurements were conducted on a universal tensile testing machine (WDW-05, Jinan Shidai Group Co., Ltd., China) with a 500 N load cell at room temperature. The compression tests of the sponges with dimensions of 1 × 1 × 1 cm3 were performed with a strain rate of 50 mm min-1. There was no interval between each cycle for cyclic loading tests.

Relevant statements have been revised in the Revised Manuscript, please refer to Page 4 Line 155-159 of the Revised Manuscript.

Q5. Is there also an option, that I misunderstood the idea of Authors. However, at lest the Characterization part has to be described in more details to make all procedures clear. If yes, I can agree to modified the decision to "minor revision"

Response: We thank the reviewer for the valuable comment and we have revised the Characterization part for a better description.  Relevant statements have been revised in the revised manuscript, please refer to Page 4 Line 155-159 of the Revised Manuscript.
